# Using Virtual Reality to Stimulate Healthy and Environmentally Friendly Food Consumption among Children: An Interview Study

**DOI:** 10.3390/ijerph18031088

**Published:** 2021-01-26

**Authors:** Eline Suzanne Smit, Marijn Hendrika Catharina Meijers, Laura Nynke van der Laan

**Affiliations:** 1Department of Communication Science, Amsterdam School of Communication Research/ASCoR, University of Amsterdam, PO Box 15791, 1001 NG Amsterdam, The Netherlands; M.H.C.Meijers@uva.nl; 2Department of Communication and Cognition, Tilburg University, PO Box 90153, 5000 LE Tilburg, The Netherlands; L.N.vdLaan@tilburguniversity.edu

**Keywords:** virtual reality, health, sustainability, food consumption, qualitative, interviews, children, environment

## Abstract

Since habits formed during childhood are predictive of adult behaviour, children form an important target group when it comes to improving healthy and environmentally friendly food consumption. To explore the potential of immersive virtual reality (VR) in this respect, we conducted a semi-structured interview study (*N* = 22) among children aged 6–13 years. This study consisted of two parts: (1) a VR experience and (2) a semi-structured interview to investigate (1) to what extent children are able to recall and understand information about the impact of food products on their health and the environment when provided to them as pop-ups in a VR supermarket; (2) what rational and emotional processes are triggered by this information; and (3) what children’s expectations about the real-life application and impact of the pop-ups are, and why. Interview data were analysed using the framework method. Results showed that although all participants were able to recall the information, only children from an average age of ten years old also understood the information. When participants understood the information, they were often aware of and felt sorry for their negative behavioural impact. Most participants expected their behaviour to positively change when imagining real-life application of the pop-ups.

## 1. Introduction

Health and environmental issues are two of the most pressing issues of our time. To illustrate, non-communicable diseases, e.g., cancer, cardiovascular diseases, respiratory illness, and diabetes, have been on the rise for decades and, in 2012, were responsible for 68% of all deaths globally [1]. At the same time, climate change, deforestation, and pollution pose serious threats to our environment [2,3]. Since both health and environmental problems are most often rooted in human behaviour, changing people’s behaviours could minimise both types of problems [4,5]. Finding ways to encourage healthier and more environmentally friendly behaviours is therefore of utmost importance. One area of behaviour that holds promise for targeting both health and environmental problems simultaneously is the consumption of food. An unhealthy diet, including the consumption of too much saturated fats and not consuming enough fruit and vegetables, largely contributes to a variety of non-communicable diseases [6,7]. Moreover, food production and consumption represent one of the largest contributors to environmental issues [8] and recent research has found that environmentally friendly food choices are generally also healthy choices [9].

When it comes to improving healthy and environmentally friendly food consumption, children form an important target group as food preferences are at least partly learned during childhood [10,11]. Indeed, earlier research has shown that eating habits that exist in childhood may track into later childhood and adulthood [12] and that children who are overweight—a potential health consequence of an unhealthy dietary pattern—have a higher chance than normal-weight children to develop into overweight adults [13]. Once formed, habits can be strong and difficult to alter, which is illustrated by the findings from a study on meat consumption; meat eating habits were found to have the strongest positive influence on meat consumption, much stronger than the also significant, negative influence of reduction intentions [14]. Given their strong influence on behaviour, the formation of healthy and environmentally friendly habits is key in achieving and maintaining healthy and environmentally friendly behaviour—something that was, with regard to sustainability, also recently suggested by others [15]—and should be done as early in life as possible.

Traditional health and environmental communication efforts aimed to target children may, however, either target their parents instead [16] or not necessarily match children’s needs, as they use non-interactive media channels to distribute the message, e.g., product packaging [17]. To elaborate, play is regarded as a core activity in a child’s development, and exploration and interaction with the environment are thought to be crucial for learning [18] as they increase engagement and as such facilitate learning [19]. Compared with traditional communication channels, immersive VR may prove to be more effective for children to learn about the healthiness and sustainability of food than traditional approaches, as it provides them an opportunity to freely explore an environment, interact with it, and become immersed in it—in other words, to play with it. Virtual environments are computer-generated 3D-models in which participants can experience, and with which they can interact intuitively in real time [20]. VR is a “reality” designed to simulate a person’s physical presence in a specific environment, and it has been shown that subjects quickly feel “present” in such VR environments, such that the actual situation is suppressed in favour of the virtual situation [21]. VR has been successfully applied in a wide range of fields, including medicine (e.g., surgery training [22]) and learning traffic rules [23]. There is abundant evidence of the relevance of play in learning and, more generally, healthy child development. It has been shown that play enhances children’s learning readiness [24] and problem-solving skills [25]. While playing, children explore the world, and they discover new competencies and build self-confidence [26]. Moreover, in digital media, what contributes to engagement and immersion is when users physically interact with the medium and when there is a natural mapping [27], such is the case with immersive VR in which natural physical interactions are possible via a head-mounted display.

To date, however, not much is known about the effectiveness and user experience of VR applications that aim to educate children about health and sustainability. A recent study compared a traditional learning approach (reading-based learning) with a VR-based learning system in which children aged 7–9 years old were exposed to animations and posters depicting information about healthy food [28]. This study found that VR improved the learning performance and that children enjoyed learning through VR more than through the traditional learning approach. Furthermore, a small-scale study among high-school students showed promising effects of an immersive VR experience on knowledge about ocean acidification [29]. Lastly, some preliminary data [30] showed a tendency of healthier food choices after a VR experience that taught children aged 8–10 years about the healthiness of foods. At the same time, especially young children’s ideas about food are often not rational and instead based on perceptions of a product’s attributes, e.g., the appearance of the product or its package [17]. This is related to their reasoning capabilities not yet being fully developed and the resulting difficulty for young children to choose food based on conceptual motives [31], e.g., health or environment. To truly understand how VR might educate children about the healthiness and sustainability of their food choices, it is thus important to not only explore their rational responses to the VR experience, but to also especially enquire about their less rational responses.

To contribute to the scarce health communication evidence available and provide initial knowledge on this subject within the field of environmental communication, we conducted a semi-structured interview study among children (i.e., aged under 18 years) with the aim to obtain an answer to the following research questions (RQs): (1) To what extent are children able to recall and understand information (text + visual) about the impact of food products on their health and the environment when provided to them as pop-ups in a VR supermarket?; (2) What rational and emotional processes may be triggered by this information?; and (3) What are children’s expectations about the real-life application and impact of the pop-ups, and what may be reasons for these expectations?

## 2. Materials and Methods

### 2.1. Design and Participants

This was a qualitative interview study, for which data were collected through semi-structured interviews with attendees of the Weekend of Science event. This event is organised annually, by commission of the Dutch ministry of Education, Culture, and Science, and intends to provide everyone—young and old—with an accessible insight into the world of science and technology. The event is hosted at different locations such as museums and universities (more information in Dutch can be obtained from https://www.weekendvandewetenschap.nl).

In total, 25 interviews were conducted in one day, i.e., on 5 October 2019, on one of the campuses of a large Dutch university, in a room especially dedicated to this study and based on a walk-in procedure (convenience sampling method). Three interviews (12%) had to be excluded from the analysis, as participants did not fulfil the inclusion criterion of being a child (i.e., under the age of 18)—participation in the VR experience by several adults was possible, given the nature of the event. This resulted in a final sample of 22 participants. Participating children were on average 9 years old (range 6–13), and more than half of them were female (n = 13, 59.1%). All participants had the Dutch nationality and were fluent in Dutch, except for one girl who was Belgian and spoke Belgian Dutch.

### 2.2. Procedure

The study was approved by the ethics committee of the University of Amsterdam (reference number: 2019-PC-11168). All participants received a brief explanation about the study aims and procedures, as well as information about their rights and the confidential handling of their data. After participants and their parents provided their written informed consent, they were able to take part in the study, which consisted of two parts: (1) a VR experience and (2) a semi-structured interview about this experience. The immersive VR supermarket employed in the VR experience, the immersive VirtuMart [32,33] was designed by one of the authors of this manuscript (i.e., L.N.v.d.L.) and adapted for the current and a related [34] VR study.

The VR supermarket experience started with a short instruction on the VR equipment (i.e., headset and controllers) and an explanation that during the VR experience participants would be able to walk freely within the dedicated area that we marked in the room where the study was conducted. Then, participants took a practice round, in which they were asked to navigate through the VR supermarket, pick up a product with their virtual hands and place it into a shopping basket. After successfully completing this practice round, participants continued to the next part of the VR experience, where they were verbally instructed to find the shelves with fruit biscuits, in the supermarket. Participants were informed that there were six options to choose from within this product category, and that they ultimately were able to pick one option that they would like to buy had they been in a real supermarket. Before being able to make a final selection, participants were told that they had to pick up each option because by picking up the product, a pop-up with information about the presence of palm oil in the product would automatically appear. The information was given both textually and with an accompanying image or, in other words, a visual-impact message. In earlier research, visual-impact messages have been found to positively influence recycling attitudes and intentions [35]. The pop-up either had a health appeal (n = 9)—where information focused on how palm oil can contribute to an unhealthy weight [36]—or an environmental appeal (n = 13)—with information appearing on how palm oil can destroy the rain forest [37]. See Figure 1 and Figure 2 for an example of a pop-up with a health appeal and environmental appeal, respectively. Within the product category of fruit biscuits, two options posed no risk, two a medium risk, and two a high risk of the consequence of consumption depicted in the pop-up. Participants were instructed to carefully look at each pop-up, before making their final decision for a product. They ultimately had to place the product of their choice in one of the shopping baskets. In our study, the children picked up the products with handheld controllers and were able to physically walk around in the environment, contributing to their immersion in the virtual environment. Furthermore, the pop-ups really only popped up when children pick up a product of their choice, making it different from a non-interactive brochure just lying next to the product.

Immediately after completing the tasks in the VR supermarket, the participants were asked to participate in the interview. At the start of the interview, participants were briefly reminded of the purpose of the interview, i.e., to get more insight into participants’ ideas and opinions regarding the pop-ups they encountered during the VR experience. Interviews were audio-recorded and lasted for 158–237 s (*M* = 200 s). 

### 2.3. Apparatus

The VirtuMart was designed in Blender/Unity3D. We used an HTC Vive set, consisting of one VR headset, two base stations, and two controllers. The two controllers simulated the participants’ hands, which ensured the VR experience would feel as realistic as possible, and enabled participants to pick up products and teleport (i.e., navigate) through the supermarket. The base stations enabled the participants to walk through the supermarket—as long as they stayed within the area that was covered by the base stations, an area marked on the floor for participants to notice where they could (not) walk. Unity (version 5.6.0f3) was used to run the VR application.

### 2.4. The Interview Guide

A semi-structured interview guide was developed, that included open-ended questions related to the three research questions of this study. First (RQ1), a question was asked to shed light on two important prerequisites for behaviour change according to the elaboration likelihood model [38,39], i.e., recall and understanding of the message that was conveyed through the pop-ups; “Could you tell us, in your own words, what you saw on the pop-ups?”. Second (RQ2), a question was asked to increase our understanding of any emotional and rational processes that may have been triggered by seeing the pop-ups, i.e., “What kind of feelings and thoughts did these pop-ups evoke in you?”. Third (RQ3), the interview guide included questions on participants’ expected evaluation (“What would you think of seeing these pop-ups in the real supermarket or in the school or sports canteen?”) and behavioural impact (“To what extent do you think you would be affected by these pop-ups?”) of the pop-ups in real life, as well as any reasons for these expectations. The interview concluded with the assessment of several demographic variables. The complete interview guide can be found on the Open Science Framework (https://osf.io/5ktzx/).

### 2.5. Data Analysis

Data analysis was conducted using the framework method [40]. First, interviews were transcribed verbatim—interview transcriptions can be found on the Open Science Framework (https://osf.io/5ktzx/). Second, two coders (E.S.S and M.H.C.M.) familiarised themselves with the interview content by reading all transcripts. Subsequently, they developed a code book based on which they independently analysed the same single transcript, after which any inconsistencies were discussed and resolved, and one additional code was added to the code book. In a second round of coding, both coders independently coded the same three subsequent transcripts, again followed up by an open discussion and the addition of another code to the code book. In a third round, the same procedure of independently coding three transcripts was used, after which it was decided that the code book was now final. This final code book (which can be found in Appendix A), consisted of multiple major and minor themes for every main interview question and was perceived to cover all relevant information. 

With the final code book, the remaining 16 transcripts were independently coded by both coders, after which the percent agreement was assessed. Percent agreement reflects the degree of similarity between coders in assigning the same code to the same piece of text (i.e., percent agreement = the number of coding agreements divided by the number of agreements and disagreements combined [41]), with a coefficient of ≥90% being considered acceptable [42]. It was determined that the intercoder agreement was sufficiently high, namely 91%. Nonetheless, coding inconsistencies were again discussed and resolved. With all transcripts coded, themes were grouped together in clusters, and data clusters were interpreted by looking for patterns and identifying answers to the research questions.

## 3. Results

### 3.1. Pop-Up Message Content Was Well Recalled, but Understood Only from a Certain Age Onwards

When asked to describe, in their own words, what participants had seen on the pop-ups in the VR supermarket, all participants were able to recall (at least one of) the images they were presented with on the pop-ups. Nonetheless, based on the answers given, we deduced that not all participants understood what message the pop-ups aimed to convey, i.e., the impact the consumption of the different food products would have on their health or the environment. While 15 participants understood the information provided, one participant only understood the message after she asked the researchers to read the information out loud for her during the VR experience, and six participants did not understand the information.

“Well, the girl had black hair with a purple tail, and I don’t remember what her shoes looked like, or that she wasn’t wearing shoes. I don’t remember that”(Dutch girl, 8 years old).

“I’ve seen a lot of monkeys. Orangutans. Or at least, brown monkeys. And then it said: ‘this product is good or bad for the rainforest’”(Dutch girl, 10 years old).

Interestingly, the participants who understood the message were on average almost ten years old (*M* = 9.93; range = 6–13), whereas the participants who did not (directly) understand the message were on average younger than eight years old (*M* = 7.71; range = 7–9).

### 3.2. With Understanding Came Rational and Especially Negative Emotional Responses

When asked to describe the thoughts and feelings that arose as a result of seeing the pop-ups, only half of the participants mentioned specific thoughts and less than half of the children mentioned positive and/or negative feelings as a response to this question—for the remaining participants, no clear answer could be obtained. Of the children that did provide an answer to this question that either contained thoughts or feelings (n = 14), more participants reported on thoughts than on feelings, with the children reporting on thoughts being slightly older (*M* = 9.82; range = 7–13) than those reporting on feelings (*M* = 9.44; range = 8–13).

Of those that reported on thoughts (n = 11), all but one participant indicated more awareness of their behavioural impact after seeing the pop-ups—all of these participants also correctly understood the message. This one participant reported wonderment as a thought—this was the only participant reporting on thoughts, while not understanding the message.

“Well, I thought, yes these are healthy, and these are not”(Dutch boy, 11 years old).

“With one I thought this is bad and with the other I thought this is good. [Interviewer: And why did you think that?] Because if the jungle breaks down, I will no longer have any jungle”(Dutch boy, 12 years old).

Of participants who reported on feelings (n = 9), slightly more respondents reported on negative than on positive feelings, with feeling sorry being mentioned most often and guilt being mentioned once—both negative feelings, however, pertained to environmental information only and were not mentioned by participants that received pop-ups with a health appeal. All participants who reported negative feelings understood the message. The single positive feeling mentioned was a general liking of the pop-ups, and this feeling was mentioned almost as often by participants who did and by participants who did not understand the message, and both by participants who were presented with pop-ups with health information and participants who received environmental information. 

“I thought they [i.e., the dolls] were cute”(Dutch girl, 8 years old).

“Well, that I wouldn’t buy that product because I feel sorry for the monkeys. They live there too. And then <laughs> you actually demolish their house, and I don’t think you can do that”(Dutch girl, 13 years old).

### 3.3. Mixed Expectations about Encountering Pop-Ups in Real Life, but the Expected Behavioural Impact Is Positive

When asked how participants would evaluate seeing these pop-ups in the real supermarket or in the school or sports canteen, as many participants responded positively (n = 9) as there were participants who provided a neutral response (n = 9). Only a small minority would not like seeing the pop-ups in real-life and thus responded negatively (n = 3). Two out of the three participants with a negative evaluation simultaneously also reported a positive evaluation, though—indicating mixed expectations.

When asked to elaborate on their response, participants who expected to positively evaluate the pop-ups, most indicated this might be because of the pop-up increasing awareness (n = 6), and some suggested the pop-up might have another (additional) positive influence (n = 3), e.g., on their own or others’ consumption behaviour. All of these participants correctly understood the message the pop-ups aimed to convey. A minority of two participants indicated the likability of the visual part of the pop-up as a reason for the positive evaluation—yet only one of these respondents also understood the message.

“Yes good. [Interviewer: And why?] Well, because then you know that this [product] can make you very fat and that [product] not”(Dutch boy, 11 years old).

“Good. [Interviewer: Good?] I do think that far fewer people then just buy [products]. [Interviewer: And why do you think that?] Well, because people end up like: ‘Yes, what I’m buying now is just bad’. And if it is not sold then it is no longer made”(Dutch girl, 11 years old).

“Cute too. Because monkeys are cute”(Dutch girl, 10 years old).

Only one of the participants with a negative evaluation provided a reason for this, which related to the peculiarity of the pop-up in real life—this was also the single reason given by almost all participants who reported a neutral evaluation expectation (n = 8).

“Weird. [Interviewer: Why weird?] Ehm, because then suddenly out of nowhere I would encounter a picture with a monkey …”(Dutch girl, 8 years old).

When asked to what extent participants believed they would be affected by the pop-ups in a real-life setting, the large majority of participants (n = 16) indicated that their decision would indeed be influenced by the pop-up, whereas a minority of participants indicated this would not be the case (n = 6). The main reason for an expected impact of the pop-up was related to the pop-up increasing awareness of the environmental or health impact of participants’ own behaviour. All participants who provided this reason for expecting an impact of the pop-ups on their behaviour (n = 14) correctly understood the message the pop-ups conveyed, while all participants who did not explain their expected impact based on this reasoning either did not expect to be influenced at all or did not correctly understand the message (n = 2).

“Yes, I think you better take this one [i.e., the slim doll] now because it is healthier”(Dutch boy, 11 years old).

“Oh yeah. [If I see] the first one, I’ll take it, and when I see the destroyed one [i.e., the destroyed rainforest], I’ll put it back”(Dutch boy, 9 years old).

## 4. Conclusions and Discussion

### 4.1. General Discussion

This study aimed to contribute to the scarce health communication evidence and—to the best of our knowledge—lack of environmental communication evidence on the use of immersive VR to educate children about the health and environmental impact of their food consumption. To this end, we conducted semi-structured interviews with children aged 6–13 years.

With regard to our first research question, i.e., to what extent children were able to recall and understand information (text + visual) about the impact of food products on their health and the environment when provided to them as pop-ups in a VR supermarket, we conclude that this is indeed possible but only from a certain age onwards. To elaborate, participants who understood the impact message were on average almost ten years old, whereas the participants who did not understand the impact message were on average younger than eight years old. Most likely, and supported by anecdotal evidence gathered during the conduct of the study, children younger than ten years old may have difficulty reading the text that accompanies the images (in a relatively short time span). This may imply that the textual information has no or not much added value for younger children, while the images used in the present study were in themselves not sufficiently able to explain the health and environmental impact of the children’s food consumption. To circumvent this problem, future research might test the idea that the images may need to be accompanied by audio instead of text—an idea supported by the finding that just over 40% of children have been found to be able to read at the age of seven, and at the age of eight still more than 20% are unable to do so [43]. Another solution might be to enhance the VR experience by not only showing children the consequences of their food consumption using pop-ups (with images supported by either text or audio), but also having children experience these consequences themselves, e.g., by stepping into the “shoes” of the child that becomes overweight or of the orangutan whose habitat is being destroyed in a 3D scenario [44]. This potential solution should receive attention in future research efforts too. Another explanation for our finding that only children from about ten years old understood our message, might be that children of a younger age are not yet capable of processing information about abstract subjects such as health and the environment. Nevertheless, earlier research among children has shown that children as young as 8.7 years on average were very well capable of identifying high and low caloric products as, respectively, unhealthy and healthy [45]. Although this might be different for the few children in our sample that were only six or seven years old, we thus believe this might not be the sole explanation. At this moment, however, we would advise to use the pop-ups used in the present study only when targeting children aged ten years and older.

When we asked children to identify what rational and emotional processes were triggered by the information provided in the pop-ups, the most notable finding was that among participants who understood the message, the most often mentioned thought triggered by the pop-ups was an increase in awareness of their behavioural impact on both children’s own health and the environment. This finding is in line with results from previous research on visual-impact messages [35,46]. The most often mentioned feeling among participants with a good understanding of the information was feeling sorry, yet this only related to the children who were presented with information about the environmental impact of their behaviour, not to those who received health-related information. This may be a result of a difference in emotional valence of the pop-ups. That is, the environmental pop-ups displayed the consequences of children’s food choices on orangutans’ habitat—with the worst food choices being associated with sad-looking monkeys in a destroyed rainforest—while the health pop-ups showed how food choices may affect body weight—with the heaviest people still looking rather happy. For future research, it might be good to aim for more comparable stimuli in this regard or to use more neutral stimuli, e.g., not including animals and/or people [34].

Our findings may imply that the pop-ups in the present study were able to generate an increase in awareness among children about their behavioural impact, when it concerns the impact of their behaviour on their own health and when it concerns their environmental impact. As awareness can be considered an essential prerequisite for behavioural change [47], this is an important finding. However, before drawing any firm conclusions based on the findings presented here, future research should invest in finding quantitative support for this finding as well as for potential effects on other important behavioural predictors when it comes to health- and environment-related behaviour, such as efficacy beliefs [35,48,49]. Moreover, it is important to mention that, when asked, half (n = 11) and over half (n = 13) of the children were not able to indicate any thoughts or feelings, respectively. This might indicate that although children might understand the message, it might be difficult for them to explain the message and the thoughts or feelings it generates in their own words—more research into this topic might be needed to provide more insights in this regard.

With regard to the real-life application of the pop-ups, most children had neutral or positive expectations. In the case of positive expectations, children most often indicated that these were caused by the expected behavioural impact of the pop-ups. In line with this finding, the large majority of participants indicated that they expected their decision to indeed be influenced by the pop-up when imagining encountering the pop-up in real life, with the main reason for this expected influence of the pop-up being an increase in awareness of the environmental or health impact of their own behaviour. Whereas the conclusions drawn based upon this small-scale qualitative study should be interpreted with caution and ideally be replicated in larger-scale quantitative research before translating the findings into practice, the findings suggest that efforts aimed at educating children about health and the environment might benefit from using immersive VR—yet only from an average age of ten years onwards when encompassing textual information. Given that VR becomes increasingly accessible through, for instance, its reduced cost and increased compatibility with smartphones, the use of immersive VR as a playful educational tool holds promise for these children and is no longer out of reach. Moreover, the results suggest that it would be worth investigating quantitatively whether providing information on—even young—consumers’ behavioural impact on the spot has a positive influence on food choices in relation to health and sustainability issues. Even more than VR, the use of augmented reality (AR) applications holds promise in this regard as AR has the ability to connect the real with the virtual world [50]. For instance, AR might enable the scanning of a product’s barcode in a supermarket to result in information on the health or environmental impact of that specific product being presented on one’s smartphone—as such letting the pop-ups that we now studied in a virtual world appear in the real world. As young children, who form strong habits already during their childhood, will obtain increasing buying power as adolescents and are the parents of our future, it important to educate them about the health and environmental impact of their product choices as early in life as possible to be able to enhance the likelihood of a healthy and sustainable future for everyone.

### 4.2. Limitations

There are some limitations that need to be taken into account. First, the sample of children that took part in the study might not be entirely representative of the Dutch population of children in this age group. All participants took part in the study as part of their attendance at a Weekend of Science event on one of the campuses of a large Dutch university. Although this annual event aims to provide all people—of different ages and with different backgrounds—insight into the world of science and technology in an accessible way, attendees of this event were by nature interested in science and/or technology and study participants took part because of their interest in a VR experience. As such, a certain degree of selection bias is likely to have occurred. Future research efforts might therefore aim to complement the study findings with findings from contexts that may provide a slightly different perspective, e.g., by evaluating the VR experience among children from primary and/or secondary schools in more rural areas of the country. Second, our convenience sampling method prevented us from collecting data until we reached saturation, a preferred and commonly employed data collection method in qualitative research efforts [51]. Although we cannot be a hundred percent certain that we reached data saturation in our study, and future research efforts aimed to evaluate the VR experience might thus benefit from developing a recruitment strategy that would allow for data collection to continue until saturation is reached, the final dataset consisted of 22 interviews that provide a variety of perspectives from children of varying ages, and we were in fact able to identify some clear patterns in the data.

## Figures and Tables

**Figure 1 ijerph-18-01088-f001:**
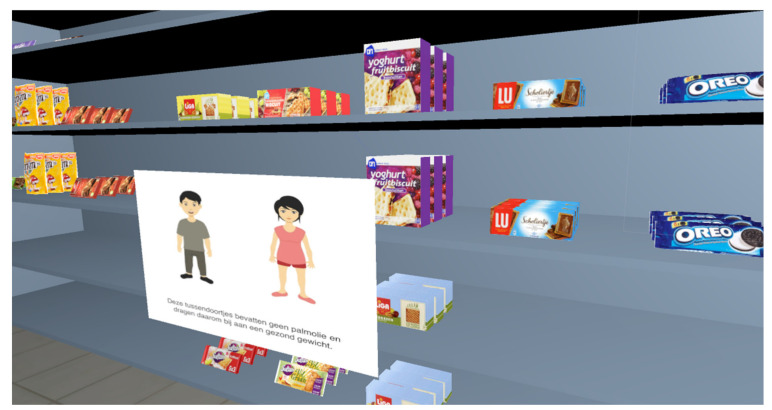
Pop-up with a health appeal (picture: Vecteezy.com).

**Figure 2 ijerph-18-01088-f002:**
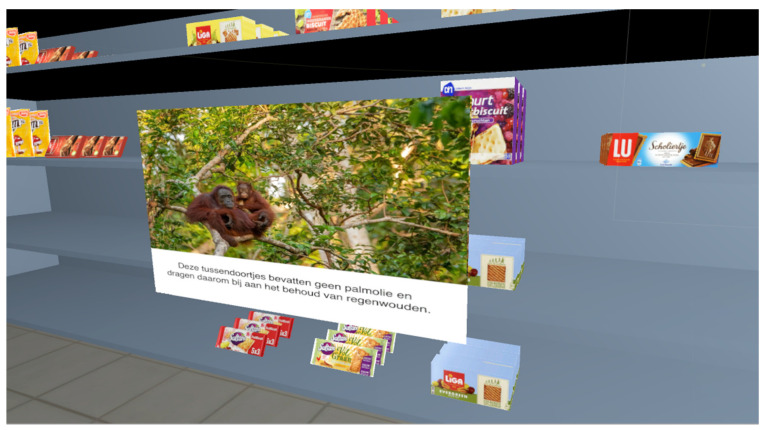
Pop-up with an environmental appeal (picture: pixabay.com).

## Data Availability

The complete interview guide and interview transcriptions can be found on the Open Science Framework (https://osf.io/5ktzx/).

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
