# Peer review of "Using Virtual Reality to Stimulate Healthy and Environmentally Friendly Food Consumption among Children: An Interview Study"

_ijerph, 2021, doi:10.3390/ijerph18031088_

Round 1
Reviewer 1 Report
This manuscript describes a qualitative study of children who shop at a Virtual Reality supermarket. The introduction provides solid justification for the twin focus on environment and health behavior through a focus on food consumption behaviors. The creative VR study design is a strength and the age comparisons are instructive about who could be a potential target of a VR attitudinal/behavioral change strategy. The discussion of ways to address lack of reading ability were thoughtful. The manuscript, however, has some areas that need attention.
The literature review would be strengthened with some evidence as to why 6-13 year olds are targeted as research finds that development of food preferences occurs earlier in life. What does the research suggest about food attitudes or behavior changes for children in the age group under study? I also wondered if everything that can be done in VR is considered play?
Line 54-56 - Children’s communication needs – Play is an activity that includes communication but I have not seen it defined as a “communication need”. Support for this label is needed or perhaps use a different term for this concept. Some discussion of receptive language or research on from who/how children learn would be helpful here, as well as the relevance of play as a learning tool for school age children.
Line 60-61 – define and cite sources for “traditional communication channels” for children
Interesting research questions but the second one is not well-connected to the lit review. Some information on what is known about rational and emotional process related to food behavior and/or pop-ups (or advertising?) in children is needed.
Line 150-152 – provide a cite for the behavior change perquisites statement
Lines 153 – 157 How were the questions for rational and emotional processes developed? How are these concepts defined? Some examples of the questions used to investigate the three research questions are needed here. I wondered why the question about behavior impact was related to seeing pop-ups in real life. I expected it to be about whether the VR pop-ups would change the children’s future food choices. Would real-life pop-ups be considered “play”? Are they feasible in stores? Some justification for the third research question is needed. [Link to AR?]
Lines 182-200 – Some quotes that support the claim that 15 participants understood the message are needed here.
Lines 201-202 – Define thoughts versus feelings either here or in the methods sections. Are thoughts equivalent to rational and feelings equivalent to emotional? This would help the reader understand why “cute” categorized as emotional but “good” or “wonderment” is categorized as thought.
Lines 229-231 – This information would be better placed in the paragraph that starts this section (Lines 201-206). How many altogether mentioned either a thought or a feeling? Were there any age differences in who talked about emotions versus feelings versus nothing?
Line 252 – To me, this quote doesn’t sound like the girl correctly understood the pop-up.
Lines 255-260 – This paragraph was confusing as positive, negative, and neutral responses are combined. Please consider either separating the ‘positive because of likeability’ into its own paragraph with a quote (Lines 255-256) or including that theme in the earlier discussion of positive evaluations without the quote.
Lines 310-329 A discussion of why feelings were triggered only by the environmental pop-ups would be interesting. Are feelings an important component in the behavioral change model?
Lines 335-338 Some discussion of how your sample (children attending a science event at a university in an urban area) may influence the findings is needed. What might this say about their reading levels and comprehension, and familiarity with VR relative to other children?
Lines 335 – The statement about the benefits of VR and its positive impact on child behavior go beyond the data provided here. The study conclusions should be limited to the new hypotheses that are raised about VR, for example, that older children may be the best target, that non-textual VR pop-ups should be considered, and that environmental concerns produce stronger feelings than health concerns. There is no evidence of effect. The discussion of AR is interesting – it should come earlier as a justification for the final research question.
Reviewer 2 Report
Abstract:
While I understand providing the aims are important, I did not quite understand the methods of this study through the abstract. I would recommend removing the aims to provide more on methods, specifically analysis.
Methods:
Include the type of study this is and sampling strategy (it sounds like convenience).
What, if any, was the underlying theory driving this research? Specifically the qualitative analysis?
Conclusion/Discussion:
There are many components missing from the Discussion portion. For example, where do the authors tie their findings to current research specifically around behavior change and health communications. T
Most of the Conclusion/Discussion unpacks the results, which is good, but again, does not tell the reader the implications of these results.
Most importantly, there are no limitation mentioned in this study. With a small sample size, convenience sampling, etc. there is much to explain in regards to the limits of these findings.
Another important element is that while the authors mention how VR is becoming more readily available, there is no evidence for this. As this study indicates, participants only had access because they attended an event, not because it was available in school or at home.
Finally, there is no mention of what the overall intention of this proposed intervention in the long term. If young children are being targeted, we know from previous nutritional studies that without approval from the parents, children will not have access to these nutritious foods.
Overall this is an interesting study, but lacks foundational behavior change components around nutrition.
Reviewer 3 Report
The study is interesting in that it considers two contemporary issues of public health/nutrition and environmental impacts. Additionally, the virtual reality experience and youths' insights are potentially interesting to education and communication researchers and practitioners. Some feedback:
Introduction:
For claims such as lines 54-46 - include references to support claims; also make sure to clarify how pop-ups in a VR simulation are more meaningful/immersive than a brochure; it seems pop-ups in a VR simulation still require reading and one-way engagement communication, much like a brochure or book
Line 60 - when VR is first introduced, provide more background information and details about what it is, how it's created, and how it is used in science communication and education
Line 76 - research question one is currently a yes/no question - consider revising it to more broadly reflect the phenomenological qualitative methods used about participants' experiences
The authors might consider also including some information in the literature review about message framing in communication, since the research examined impacts of the message frames in pop-up text.
Methods:
before section 2.0, include some discussion of methodology - is this a phenomenological case study? ontology/epistemology of the study? researchers' roles and biases?
Results:
The results still appear to be reported as chunks or categories from the qualitative coding. Consider reporting the results are themes. For instance, research question/result 1 - theme: Youth were able to recall and describe pop-up message content from the simulation.
Discussion and Conclusions:
While the study was able to examine the impacts of the text pop-ups, are there any recommendations for future VR research that can be made to evaluate the entire VR experience and not only the pop-up text windows?
